# The Impact of an Agent's Voice in Psychological Counseling: Session Evaluation and Counselor Rating

**Hyo Chang Kim, Min Chul Cha and Yong Gu Ji***

Department of Industrial Engineering, Yonsei University, Seoul 03722, Korea; huychag1@yonsei.ac.kr (H.C.K.); shinyfe74@yonsei.ac.kr (M.C.C.)
* Correspondence: yongguji@yonsei.ac.kr; Tel.: +82-2-2123-7836

**Abstract:** As an agent delivers content during the communication between an artificial intelligence (AI) machine and a person, the voice of the agent is a crucial factor to be considered. Particularly in the fields of eHealth, the perception of users toward an agent is crucial as it significantly affects the communication between the agent and its patients, as well as the treatment results. Thus, this study examined the effects of the voice of an agent on the perception of users toward the agent and its counseling effects. This study developed a psychological counseling agent with four voices according to gender and age, communicated with the subjects through such agent, and measured the perception of users toward the agent and its counseling effects through a questionnaire. Results demonstrated that the female-voiced agent had a higher level of attractiveness than the male-voiced agent, regardless of the age of such voice, and the agent using an older voice had a higher level of expertness and depth than the agent using a younger voice, regardless of the gender of such voice. The findings of this study are expected to be effectively used to design a voice-based AI agent that considers the optimal voice according to the purpose of use.

**Keywords:** AI; voice; HRI; HAI; psychotherapy; counseling

## 1. Introduction

The emergence of artificial intelligence (AI) has diversified communication methods between computers and humans. Gunkel [1] stated that a need for a paradigm shift from computer-mediated communication to human-machine communication arose with the advent of intelligent machines. In accordance with the advancement of AI, an increasing number of discussions have been recently conducted to apply AI technology in psychological counseling. A psychological counseling system using an AI-based conversational agent (CA) can provide various solutions for overcoming the limitations of existing counseling systems. Counseling systems based on a CA do not suffer from fatigue and emotional exhaustion, etc., like that of a human counselor. People also found it easier to show their emotions objectively with less fear and pressure when communicating with a CA [2,3]. Thus, AI application in psychological counseling can be said to be an adequate field of study in the sense that AI development aims to achieve automated interactions through human-machine communication [4].

Voice provides the listener with several information about the speaker, such as its gender [5], emotional conditions, and personality [6]. It is an essential medium for human communication and a social interaction technique that people use to determine the impression and likeability of another person [7]. Upon hearing of an unacquainted voice, people automatically form the first impression of the personality of the speaker and infer traits such as trustworthiness, attractiveness, etc., of the said speaker [8–11]. Such first impressions based on the speaker's voice are formed very swiftly and remain consistent

[10,12]. Tamagawa [13] found that the perception of users toward an agent varied according to the intonation of such agent. Likewise, the voice of the voice-based CA which delivers a message plays an extremely significant role in building a relationship between the agent and its user in a human-AI communication.

Clinical counseling research shows that the effectiveness of counseling is dependent on the counselor rather than on theoretical differences, and the characteristics of the counselor is a significant element of clinical counseling therapies [14,15]. Beutler [16], performing a meta-analysis on 141 studies, addressed in his study the characteristics of the counselor which affect the outcome of counseling that observable traits of the counselor such as gender, age, and ethnicity affect the outcome of counseling. Particularly, people automatically infer traits of the agent based on its voice and such inferred traits of the agent affect the attitude and contents of the conversation between the agent and its users [17–19]. People arbitrarily interpret the persona of the CA based on its voice and such interpretation of persona affect their response to and perception of the CA [20]. Therefore, the perception of users towards the agent is crucial in the fields of eHealth, including psychological counseling [21]. As the voice of the agent exerts considerable effects on the perception of users, it should be designed with full consideration of social aspects and conversational context. However, research on human-robot interaction (HRI) related to the voice of the agent has received less attention than that on functions of robots and improvement in their linguistic abilities. Thus, this study examined the effects of the agent's voice on the perception of users toward the agent and the counseling effects.

## 2. Literature Review

### 2.1. Human Machine Communication

Machines have served as channels for communication between people for a long time. However, recent technological development has enabled intelligent machines to directly communicate with people beyond their previous role of only supporting communication between people. Human-machine communication (HMC) is defined as an ongoing sense-making process between people and machines with the changing role of intelligent machines [22]. It includes communication with digital interlocutors, such as intelligent machines equipped with conversation functions and AI agent [23]. Dr. Guzman stated that "HMC can be understood as a research umbrella encompassing… approaches to the study of technologies designed as communicative subjects" [24] (p. 7). HMC is distinguished from other fields in that it directly focuses on the communication between people and machines to increase the level of understanding between them [24]. As the role of machines in communication changes, researchers have raised questions on defining a boundary between people and machines in terms of HMC beyond a technical perspective and how to consider this boundary with respect to communication [25]. In the following section, details of CAs are examined to better understand systems designed to facilitate communication between people and computers.

### 2.2. Conversational Agent

A CA is a computer system designed to communicate with people in natural languages [26–28]. Most CAs are designed based on similar technology, whereas the application of these CAs varies significantly according to their purposes [27]. CAs and users interact with each other through voice- or text-based conversations [29]. Text-based CAs are also called chatbots, among which is the first computer program "ELIZA" developed by modeling the human language use [30]. Chatbots often provided on websites or messenger platforms for customer service [31,32]. Voice-based CAs began to be studied in the late 1980s and were implemented in the 1990s to automate self-service and call routing [29]. Thus, the initial-stage CA technology was significant in terms of enabling computers to deliver information to people in human languages. Recently, the linguistic abilities of CAs were rapidly enhanced through the development of AI technology. Additionally,

language-based interface has become more intellectualized and recognized in a social context [33], thereby broadening the use of CAs under these circumstances. Therefore, the advancement of voice-based CAs indicates a paradigm shift to CAs that both provide health care to users and serve as their intelligent personal assistants, such as smartphones and smart speakers [34,35]. CAs, including Alexa and Siri, communicate with users using voice and natural languages and pay attention to the feedback from users in conversations. These CAs also attempt to take constant follow-up communicative actions according to the needs of users. Brave et al. [7] reported that people considered the agent that showed empathic responses to them as more thoughtful, likable, and reliable than the agent that did not. CAs should be able to perform social behaviors and show emotional expressions [36,37]. People tended to personify CAs during interactions with them and preferred CAs with voices and emotional expressions abilities similar to humans over those without [38,39]. As indicated in this finding, people expect human characteristics from CAs, and their perception toward CAs will vary according to the conversation characteristics of CAs. Particularly, voice is a significantly crucial factor to be considered among various conversation characteristics of CAs, given that the entire interactions between users and AI systems are conducted through the voice of CAs. The following section reviews the characteristics of voice to better understand the effects of voice as a conversation characteristic.

### 2.3. Voice Cue

Voice provides information about the speaker, such as age, gender, and identity [40]. It not only delivers characteristics of the speaker, including gender and personality, to the listener but also makes the listener imagine the appearance of the speaker merely by hearing the voice for the first time [41]. As people develop, they can discover highly relevant clues in terms of social aspects by considering the voice characteristics of speakers, inferring gender, personality, and emotions of speakers accordingly. Voice also conveys message through adjusting its duration, changing the height of the pitch and/or altering loudness. Raymond [42] found that a fast speaking rate, when compared to a slow speaking rate, is perceived to be more persuasive, and raises positive impression on the level of competence and social attractiveness of the speaker. Mehrabian and Williams [43] also addressed that a message is perceived to be more persuasive when delivered at a faster speaking rate rather than a slower speaking rate.

Moreover, people grant humanistic characteristics to computers that speak with voice and communicate with these computers as if they are interacting with other people [7]. Previous studies on voice-based interfaces found that voice characteristics exerted various effects on people. Atkinson, Mayer, and Merrill [44] reported that the advantages of a conversation based on voice instead of text disappeared when the quality of the voice of the agent did not maintain that of the human voice in online learning environments. Qiu and Benbasat [45] mentioned that the agent that performed conversations with human voice in online shopping environments had significant effects on the perception of users toward the social presence of the agent and that such voice-based conversation increased the level of their intention of using the agent to make purchase decisions.

In HRI, defined as "the study of humans, robots, and the ways they influence each other" [46], social and cognitive interactions between a human and a robot is an essential factor [47]. However, most existing HRI studies concentrated on visual aspects of machines, such as appearances and designs, whereas research on the voice of machines received less attention [48]. A study found that the effects of the voice of machines varied depending on gender stereotypes of people. For example, computers using a male voice for conversations exerted more significant effects on the decisions of users than computers using a female voice. In addition, people were more likely to conform when the gender of computers was consistent with their own gender [49]. Nomura [50] granted gender characteristics to the voice of a robot through simple manipulation in an experiment and verified that the behaviors and emotions of people significantly varied according to the application

of gender characteristics in the voice of the robot. Edward [51] conducted extended research on the voice in HRI by applying the social identity theory. Older students in the research rated reliability and social presence of an AI system with an older voice much highly than younger students. Thus, people recognize computers and agents as social entities and become significantly affected by their voice characteristics. Particularly, people automatically infer traits of the agent based on its voice and such inferred traits of the agent affect the attitude and contents of the conversation between the agent and its users [17–20]. The following section reviews the role of the counselor in a psychological counseling and what influence the perception of the counselee on the counselor exerts on the outcome of the counseling.

### 2.4. Counselor in a Psychological Counseling

Psychological counseling is an act in which the counselor becomes a therapeutic instrument to assist the client who pleads difficulties. Studies in the past has mainly focused on the differences in the effectiveness of counseling under different counseling techniques based on specific theories. However, recent studies rather exert a growing interest and focus on the effects of the characteristics of the counselor on the outcome of such counseling [15]. Wampold [15] concluded after analyzing various studies that the key factor in the performance of the counseling is the question of which counselor conducted such counseling rather than a specific treatment used in the said counseling. Beutler [52], after analyzing 141 quantitative studies, categorized the factors of the counselor variable, which affects the performance of the counseling into the observable traits, such as gender and age, the observable states, such as the proficiency of the counselor, type of intervention, etc., the inferred traits, such as values, beliefs, attitudes, etc., of the counselor, and the inferred states from the theoretical background. Particularly, the observable traits such as gender and age has consistently been studied as the variable affecting the performance of the psychological counseling. Marecek [53] confirmed that female counselors have more conversation with the patient and express more emotions than male counselors. Buczek [54] explained the tendency of female counselors recalling more client information than male counselors. These differences can be explained in terms of socialization of gender roles. While males tend to be strong, emotionally unexpressed and refrain from revealing their emotions to others, females tend to orient towards relationships and promote self-disclosures and intimacies [55]. Studies with respect to the age of the counselor has rather not been conducted broadly because it is confounded by counselor's experience and cohort effects that reflect changes in training [56]. However, Beck [57] found that the performance of the counseling with counselors who are 10 years or more younger than the patient was comparably poorer outcomes than that of the counseling with elderly counselors.

Problem-solving through psychological counseling has depended on the patient, not the counselor. Thus, it is very critical how the patient perceives the counselor [58]. Banikiotes [59] showed that patients perceived female counselors to be more trustworthy than male counselors. Tytti [60] confirmed that a female counselor showed more positive counseling performance then a male counselor and that patients preferred to have counseling with female counselors. As such, it can be confirmed that the gender of the counselor affect the role of the counselor and the perception of the patient [61].

Thus, the gender, the age, and the perception of the counselor by the patient significantly affect the counseling effect [62]. It is also an important matter in studies on the agent to formulate positive long-term relationship with people through social interactions [63]. Therefore, by considering the gender and the age factor of the counselor and designing the voice of the psychological counseling agent, this study attempts to confirm whether there exists any difference in the performance of the counseling based on the perception of the patient on the agent through the voice of such agent. In order to identify and understand the influence of voice, psychological counseling scenarios were designed to control counseling techniques and a total of four voices were designed varying in gender and age.

### 3. Psychological Counseling Design

*3.1. Psychological Counseling Scenario*

This study established counseling procedures based on a five-staged counseling model developed by Kottler [64] and designed a psychological counseling scenario, which was formed by three psychological counseling experts to relieve the depression of users. Specifically, an initial version of the psychological counseling scenario was designed and evaluated by ten psychological counseling experts. Through a process of scenario revision according to feedback provided by the experts, the final scenario was formed as described below.

#### 3.1.1. Assessment

Basic information on counselees, such as their names and age, was obtained to structuralize personnel information and counseling. Counselees were asked to perform a depression diagnosis questionnaire, called Patient Health Questionnaire-9 (PHQ-9), to diagnose their conditions. The scores of nine items were measured on a scale ranging from 0 (not at all) to 3 (nearly every day), and these scores were added to derive the final score. A degree of depression of counselees was determined based on the scores and classified into five stages, including none (0–4), mild (5–9), moderate (10–19), severe (20–27), high risk of suicide (answer to the risk of suicidality). Those who received scores corresponding to severe and high risk of suicide stages were excluded from the experiment and guided to professional counseling institutes.

#### 3.1.2. Exploration

To confirm and specify problematic areas related to a feeling of depression, counselees were provided with six categories (i.e., sleeping, appetite, negative emotions, interpersonal relations, studying and career, and physical discomfort). Accordingly, they selected multiple problematic areas that they were facing. Then, they were asked to describe the details of their ongoing problems in a descriptive form. Through this step, they were offered with an opportunity to explain their current difficulties related to these specific categories.

#### 3.1.3. Understanding

To examine the experience of counselees in handling issues related to problematic areas, they were asked to choose when these issues began within two options (that is, just recently and for a long time). Then, they were asked to freely describe their experiences in handling these issues, including methods that they used to solve these issues.

#### 3.1.4. Action

To solve such issues of counselees, three types of sub-scenarios related to information supply, advice, and directions were designed and provided to them. A sub-scenario related to information supply was developed as a leisure activity planning scenario asking counselees about their favorite leisure activities and then encouraging them to make leisure activity plans by themselves. A sub-scenario related to advice was designed as a relaxation scenario for informing counselees with a process of straining and relaxing their bodies as well as a process of relaxing their muscles. Moreover, a sub-scenario related to directions was formed as a 'cognitive defusion' scenario ordering counselees to perform certain actions to directly face ongoing issues without avoiding them, thereby releasing psychological anxiety. Counselees were provided with three types of action scenarios and asked to select one of them and take actions accordingly.

### 3.1.5. Evaluation

After counseling with the psychological counseling agent was completed, counselees evaluated their psychological counseling experience with the agent through a questionnaire provided.

### 3.2. Psychological Counseling Agent

The psychological counseling scenarios were designed in four voices with different gender and age through AI voice actor service (Figure 1, https://typecast.ai; Typecast. Seoul, Republic of Korea, accessed on 29 Nov 2020). The duration and loudness of each voice were identically designed. Subjects were randomly assigned to four groups according to the voice and were asked to answer a questionnaire containing the following questions on the voice to verify the designed voices: "What do you think is the gender of the voice?" (feminine; masculine; and gender-ambiguous), "How old do you think the voice is?". The result of the questionnaire is as the Table 1 and four types of voices were distinguished. Moreover, the voice of the agent was output through CLOVA AI speakers developed by Naver, a Korean IT company, to enable people to communicate with the agent in the same manner that they communicate with general AI speakers.

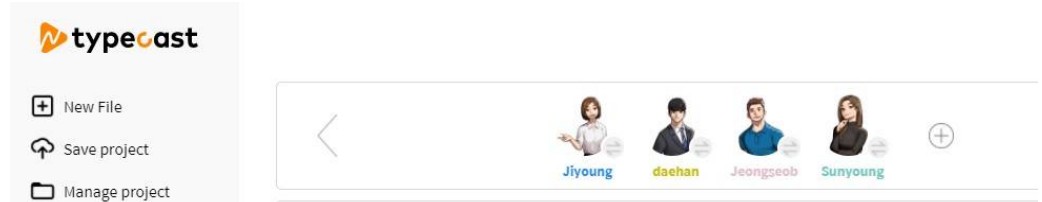

**Figure 1.** AI voice actor service "Typecast".

**Table 1**. Four types of voices.

|  | Younger | | Older | |
|---|---|---|---|---|
|  | **Male** | **Female** | **Male** | **Female** |
| Feminine | 0% (0) | 100% (22) | 0% (0) | 100% (22) |
| Masculine | 100% (22) | 0% (0) | 100% (22) | 0% (0) |
| Gender -ambiguous | 0% (0) | 0% (0) | 0% (0) | 0% (0) |
| Age M (SD) | 27.18 (4.40) | 28.09 (3.51) | 48.63 (6.79) | 47.63 (5.43) |

## 4. Method

### 4.1. Participants

Subjects were undergraduate and graduate students from the research institute that the researchers were affiliated to. The research was advertised on the university's bulletin boards and its online community. The subjects were selected from those who voluntarily expressed an intention to participate pursuant to the advertisements. Criteria for selecting the volunteers: (1) currently uses smartphones, (2) gave his/her consent for the purpose of the research and voluntarily expressed an intention to participate, and (3) had no difficulties in communicating with the researchers. Ninety-one applicants applied to participate in the experiment conducted in this study. Among them, two applicants who positively responded to the ninth question on suicide on the PHQ-9 and one applicant who did not completely respond to the questionnaire were excluded from data analysis according to the psychological counseling scenario. Consequently, 88 applicants (44 males and 44 females), whose age range was 20–30 years old (Age: 25.66, SD = 2.844), participated in the experiment. Subjects were randomly divided into four groups according to the voice types of the psychological agent. Each group was comprised of 11 male and 11 female subjects.

The research was deliberated and approved by the National Bioethics Committee (IRB No. 7001988-201901-HR-469-03, 7001988-202001-HR-787-03).

The instruction manual provided to each subject contained explanations on the participation and the possible shut down of the trial, possible side-effects and its countermeasures, and rewards for the participation. The collected data was assigned with IDs pursuant to the personal information protocols to protect the subjects' privacy. Each subject was rewarded with $20 for his/her participation in this research. Some subjects who were found inappropriate for the research were relieved of the trials and compensated with the same reward.

### 4.2. Procedure

The experiment was conducted for approximately 1 h per subject at the laboratory in the research institute to which researchers belonged. Subjects were provided with explanations on research purposes and procedures and asked to voluntarily sign an agreement of participation. Before the experiment was executed, they filled out demographic information and practiced conversation with the agent through training sessions. During the experiment, they had undergone psychological counseling with the psychological counseling agent that used different voices according to experimental groups. After the conversation was finished, subjects were asked to complete questionnaires to evaluate the level of their perception toward the agent and the level of counseling.

### 4.3. Measures

#### 4.3.1. Counselor Rating Form-Short (CRF-S)

This study used the Counselor Rating Form-Short (CRF-S) developed by Corrigan and Schmidt (1983) to measure characteristics of the psychological counseling agent recognized by subjects. This form includes 12 items based on four adjectives indicating characteristics of counselors in terms of three dimensions (that is, attractiveness, expertness, and trustworthiness). Dimension of "Attractiveness" consists of four adjectives "friendly", "likeable", "sociable" and "warm"; dimension of "Expertness" consists of "experienced", "expert", "prepared", and "skillful"; dimension of "Trustworthiness" consists of "honest", "reliable", "sincere", and "trustworthy". Conditions of subjects are evaluated based on a seven-point Likert scale. A higher total point means that a subject recognizes counselor characteristics positively. A higher score related to the three dimensions also means that a subject recognizes relevant characteristics of the counselor positively. This study examined characteristics of the psychological counseling agent perceived by subjects using scores related to these three dimensions.

#### 4.3.2. Session Evaluation Questionnaire (SEQ)

The Session Evaluation Questionnaire (SEQ) developed by Stiles (1989) has been frequently used in numerous studies to measure session and post-session effects. Therefore, this study adopted this questionnaire to analysis the results. The SEQ comprises two session evaluation dimensions, depth and smoothness, and two dimensions of subjects' post-session mood, positivity, and arousal. Depth refers to the evaluated value of counseling time, and smoothness refers to a feeling of comfort perceived during counseling or a pleasant feeling toward counseling. Positivity refers to a delightful feeling without fear or anger, and arousal refers to an energetic and exciting feeling. However, previous studies derived unstable results on positivity and arousal and failed to adopt a consistent research structure related to these dimensions. Thus, this study adopted the dimensions of depth and smoothness, which can be used for session evaluation to analyze counseling effects between the agent and subject according to the voice of the agent. The dimensions of depth and smoothness include five items and are evaluated based on a seven-point Likert scale.

### 4.4. Data Collection and Analysis

Statistical analyses were performed by using IBM SPSS Statistics version 24.0. After data cleaning, the one-way K-group multivariate analysis of variance (MANOVA) was conducted to determine any differences the effects of voice group (age and gender) on the five dependent variables of attractiveness, expertness, trustworthiness, depth, and smoothness. MANOVA was chosen because the dependent variables were related.

## 5. Results

Correlation analysis of five dependent variables was conducted, and the analytic result is as follows (Table 2). As the correlation of dependent variables was significant, MANOVA was conducted. A Box's M test was performed to verify an assumption on the invariance of covariance. The test result showed that $M = 59.836$, $F(45,17465.664) = 1.1186$, and $p = 0.184$, indicating that the assumption was appropriate. The MANOVA results showed that Wilks's $\lambda = 0.495$, $F(15, 221.246) = 4.277$, $p < 0.01$, and $\eta^2 = 0.209$, confirming that dependent variables showed a significant difference according to the voice of the psychological counseling agent. Table 3 lists the mean and the standard deviation of dependent variables according to the voice of the psychological counseling agent.

**Table 2.** Correlation among the dependent variables.

|  | **Depth** | **Smoothness** | **Attractiveness** | **Expertness** | **Trustworthiness** |
|---|---|---|---|---|---|
| Depth | - | - | - | - | - |
| Smoothness | 0.501 ** | - | - | - | - |
| Attractiveness | 0.341 ** | 0.632 ** | - | - | - |
| Expertness | 0.711 ** | 0.394 ** | 0.303 ** | - | - |
| Trustworthiness | 0.448 ** | 0.464 ** | 0.578 ** | 0.458 ** | - |

** $p < 0.01$.

Before an analysis of variance (ANOVA) of dependent variables, the Levene's test was performed to verify an ANOVA assumption on the invariance of error variance. The test result showed that the five dependent variables did not violate the assumption (depth, $F(3,84) = 0.49$, $p = 0.69$; smoothness, $F(3,84) = 1.11$, $p = 0.30$; attractiveness, $F(3,84) = 0.93$, $p = 0.43$; expertness, $F(3,84) = 1.11$, $p = 0.35$; trustworthiness, $F(3,84) = 2.49$, $p = 0.07$). The ANOVAs were significant for depth ($F(3,84) = 9.96$, $p < 0.01$, $\eta^2 = 0.262$), attractiveness ($F(3,84) = 7.06$ $p < 0.01$, $\eta^2 = 0.201$), and expertness ($F(3,84) = 8.90$ $p < 0.01$, $\eta^2 = 0.241$). The ANOVAs were not significant for smoothness ($F(3,84) = 2.28$ $p = 0.085$, $\eta^2 = 0.075$) and trustworthiness ($F(3,84) = 1.93$ $p = 0.132$, $\eta^2 = 0.064$).

**Table 3.** Means and standard deviations for the four voice on the dependent variables.

| Variable | Younger | | Older | |
|---|---|---|---|---|
|  | **Male** | **Female** | **Male** | **Female** |
|  | **M (SD)** | **M (SD)** | **M (SD)** | **M (SD)** |
| Depth | 3.51 a (0.91) | 3.79 a (1.09) | 4.63 b (0.83) | 4.80 b (0.88) |
| Smoothness | 5.16 (1.07) | 5.36 (0.91) | 5.12 (0.92) | 5.75 (0.65) |
| Attractiveness | 19.31 a (4.04) | 22.27 b (3.34) | 19.36 a (3.44) | 23.18 b (3.17) |
| Expertness | 16.04 a (4.03) | 17.00 a (4.73) | 20.59 b (3.39) | 21.18 b (3.83) |
| Trustworthiness | 20.09 (3.68) | 20.04 (4.32) | 19.91 (2.56) | 22.00 (2.54) |

Note. Means in a row with differing subscripts (a,b) are significantly different at $p < 0.05$.

A post hoc analysis of the ANOVA results of depth, attractiveness, and expertness was conducted based on Tukey's honestly significant difference (HSD) test. As for depth, subjects evaluated counseling time as more valuable when they experienced counseling with the agent using an older male or female voice compared to that using a younger male or female voice. Regarding attractiveness, subjects perceived that agent using a female

voice had a higher level of attractiveness than those using a male voice, regardless of the voice age. In terms of expertness, subjects recognized that the agent using an older male or female voice had a higher level of expertness than those using a younger male or female voice.

## 6. Discussion

This study analyzed the perception of users toward the agent and the counseling effects according to the voice of the agent. Accordingly, a psychological counseling agent was designed that performed a psychological counseling conversation with subjects in four voices according to age and gender. Based on the analytic result, it was verified that the characteristics of the psychological counseling agent and the counseling effects perceived by users varied depending on the voice of the agent. Among characteristics of the psychological counseling agent perceived by users, the characteristics related to the dimensions of attractiveness and expertness showed a significant difference according to the voice of the agent. As for the dimension of attractiveness represented by adjectives, such as warm, friendly, and likable, the female-voiced agent was found to have a higher level of attractiveness than the male-voiced agent. Moreover, a difference according to voice gender, regardless of voice age, was observed.

This result supports existing voice-based CA research results on a difference in voice effects according to voice gender. In a study conducted by Nass and Moon [65], people perceived that female-voiced computers were more sensitive than male-voiced computers. Mitchell [66] researched on responses of people according to voice gender and found that both male and female subjects found the female voice warmer than the male voice. In addition, according to traditional psychological counseling studies, it has been proved that clients tend to perceive the female counselors more emotionally than the male counselors, which allows female counselors to conduct emotional counseling more effectively com-pared to the male [59,66]. Furthermore, due to psycho-logical counseling research, it has been found that the higher the emotional communion between the counselor and the client results in an increase of information retrieved of the client [55]. Obtaining detailed and sensitive information from patients is very important for the health care of such patients [67,68]. Thus, in the field of eHealth, including psychological counseling, using a female voice rather than a male voice would be more effective, given that such CAs can help build empathy with people and promote information disclosure [69,70]. In this study, subjects perceived that the agent using an older voice had a higher level of expertness than the agent using a younger voice. As for environmental conditions, voices were established as independent variables, and the designed agent was operated consistently in the entire experimental process. Therefore, it can be analyzed that empirical stereotypes of subjects affected the result related to expertness. Edwards [71] argued that stereotypes of people can make themselves believe that older instructors have a higher level of knowledge and wisdom than younger instructors. According to psychological counseling studies, the counselors who aged more than the clients resulted in higher counseling performances, compared to the counselors who aged less than [57]. This result verified that, when delivering same contents, positive interactions between the agent and users can be derived through increasing the level of expertness perceived through the voice of the agent by users.

Regarding the counseling effects, only characteristics related to depth exhibited a significant difference according to the voice of the agent. Conforming with the result on expertness, subjects considered that the agent using an older voice had a higher level of depth than those using a younger voice. LaCrosse [72] stated that the counseling effects were determined by the perception of counselees of the counseling actions. That is, the counseling effects were affected by the degree of perception of counselees toward counselors. In particular, counselors who were considered more professional and attractive by counselees exhibited a greater influence than counselors who were not [73]. The results of this study show that the agent using an older female voice exhibited the highest level of

expertness, attractiveness, and depth perceived by subjects. As explained above, the perception of counselees toward counselors affected counseling during traditional psychological counseling, and this is verified by the observations of this study. Therefore, it can be concluded that counseling based on the agent using the older female voice can exert positive effects on the perception and the experience of counselees in the fields of eHealth, including psychological counseling. Conversely, the characteristics related to the dimensions of smoothness and trustworthiness showed an insignificant difference according to the voice of the agent. This study designed the psychological counseling agent for the experiment in which the entire subjects joined counseling under the same counseling scenario. This result implies that the characteristics related to the dimensions of smoothness and trustworthiness might be affected by psychological counseling scenarios. In this regard, such possibility in consideration of other potential factors should be examined through further research.

This study changed the voice of the agent based on age and gender, the most fundamental demographic factors, and verified a significant difference according to the voice of the agent. Despite the results of this study, the range of voice control factors and subjects must be gradually expanded in further research. When people practically communicate with each other or have counseling with counselors, they adjust the pitch of their voices as well as the speed and the interval of conversation. Furthermore, people tend to personify agents and expect human characteristics from them during their conversation with agents. Hence, an appropriate voice alternation technology should be developed and applied in the agent to satisfy such expectations and keep up with the automation of interactions through conversations. Moreover, the results of the study which showed that the agent with the older voice was perceived to have a higher level of depth and expertness over the agent with the younger voice might have been affected by age and stereotypes of subjects of this study. This study was conducted with subjects with an average of 25 years of age. The Hummert's Age Stereotypes Interactions Model [74] and conventional educational communication research asserted that the older voice may convey the traditional age stereotypes of wisdom and intelligence [71]. Although the results of this study support the results of conventional communication research [57], further experiments designed with consideration of the age aspect of subjects to sustain positive relationship between people and agents is required. Finally, in the experiment of this study, the subjects had one-time counseling sessions with the designed psychological counseling agent. Therefore, a long-term analysis instead of a short-term analysis should be conducted for more accurate verification. Communication strategies and perception can differ depending on short-term conversations and long-term conversations with the other person. In this regard, further research should be conducted to analyze HRI and HMC according to the voice of the agent in a long term.

## 7. Conclusions

This study analyzed the characteristics of the agent and the counseling results perceived by users in the fields of eHealth, including psychological counseling. Therefore, a psychological counseling scenario was designed, and a psychological counseling agent was developed, which was able to use four types of voices according to age and gender. Eighty-eight subjects communicated with the developed agent for approximately an hour and completed questionnaires on the characteristics of the agent and the counseling results perceived by them. The result of analyzing the characteristics of the agent perceived by users according to its voice indicated that the characteristics related to the dimensions of attractiveness and expertness showed a significant result. As for the counseling effects, the characteristics related to the dimension of depth showed a significant result. Regarding attractiveness, subjects considered that the female-voiced agent had a higher level of attractiveness than the male-voiced agent, regardless of voice age. Moreover, subjects evaluated that the agent using an older voice had a higher level of expertness and depth than agents using a younger voice, regardless of voice gender. In eHealth fields using an

agent, such as psychological counseling, the characteristics of the agent perceived by patients exert significant effects on the treatment results. Thus, the findings of this study are expected to be effectively used to design voice-based AI agents and provide services by applying the optimal voice, considering environments in which the agents are used. In addition, further research should be conducted to extensively consider more diverse voice alternation factors and long-term conversation aspects.

**Author Contributions:** Conceptualization, H.C.K. and M.C.C.; methodology, H.C.K. and M.C.C.; software, H.C.K.; validation, H.C.K. and M.C.C.; formal analysis, H.C.K.; investigation, H.C.K. and M.C.C.; resources, Y.G.J.; data curation, H.C.K. and M.C.C.; writing—original draft preparation, H.C.K.; writing—review and editing, Y.G.J.; visualization, H.C.K.; supervision, Y.G.J. All authors have read and agreed to the published version of the manuscript.

**Funding:** THIS research was funded by the Institute of Information & Communications Technology Planning & Evaluation (IITP) grant funded by the Korea government (MSIT) (2016-0-00562(R0124-16-0002), Emotional Intelligence Technology to Infer Human Emotion and Carry-on Dialogue Accordingly).

**Institutional Review Board Statement:** The study was conducted according to the guidelines of the Declaration of Helsinki, and approved by the Institutional Review Board of Yonsei University (IRB No. 7001988-201901-HR-469-03, 04 Jan 2019 and 7001988-202001-HR-787-03, 29 Jan 2020)

**Informed Consent Statement:** Informed consent was obtained from all subjects involved in the study.

**Conflicts of Interest:** The authors declare no conflict of interest.

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
