# Peer review of "The Impact of an Agent’s Voice in Psychological Counseling: Session Evaluation and Counselor Rating"

_applsci, doi:10.3390/app11072893_

Round 1

Reviewer 1 Report

The authors addressed and incorporated all of the feedback very well.

Introduction: With adding more information on the context of counseling, the introduction is well written and illustrates the contribution of this work.

Voice Cue: This is the only section I would say still misses some significant work in the field of CA. Even if the work does not concern the general CA discourse, the authors state a gap in the Human Robot Interaction studies which do not enough research on the perception of voice. So I ask myself, why the authors have chosen this field of research instead of addressing the gap within the HCI Voice Interaction discourse.

Counselor in psychological counseling: A very interesting and thorough description of the characteristics and their impact on the counselee relationship. This motivates the use of the variables very well.

Evaluation: Added dimensions for measurement help to understand the research design. The table and the description of the prototype explain the prototype in enough detail.

Discussion: The contribution is made clear.

Please, check some minor spelling mistakes which include missing or wrong articles. Sometimes the word should be used either in plural or concerned with articles. See here some examples:

The Impact of an Agent’s Voice in Psychological Counseling: 2 Session Evaluation and Counselor Rating or The Impact of Conversational Agents’

Line 44 “an” agent

Line 61 “an” agent or “agents”

Overall paper is now rounded and has a significant contribution to the use of voice agents in counseling and shows the importance to adapt the personality relative to their use of context.

Author Response

We attached the response file for reviewers' comments.

Reviewer 2 Report

The subject approached is interesting and important and the submitted article seems to have some improvements to answer reviewers’ comments.

However, we do understand that the submitted version of the article needs a final revision considering the following items:

  • The English written expression needs a revision. Some paragraphs are very long and very confusing. It is required to rephrase the sentences and rethink the choice of words to simplify and qualify the experience of the reader.
  • Based on the literature review, which is extensive, how can the authors justify the methodology design applied?
  • The questionnaire developed by the authors was delivered in an academic context, with young people. How this influenced the obtained results?
  • It is important to cross-check information between the answers given by the respondents and their demographic as well as social and cultural identity,
  • It is required to mention the limitation of the study, namely in the function of the characteristics of the people that answered the questionnaires.
  • The conclusions remain weak once considering the extensive literature review and the questionnaire made

Regardless of the statements described above, our final decision is to accept the article with minus corrections. This statement is supported by the argument that the required changes are based on the justification and rewritten of the presented piece of research.

Author Response

We attached the response file for reviewers' comments.

This manuscript is a resubmission of an earlier submission. The following is a list of the peer review reports and author responses from that submission.

Round 1

Reviewer 1 Report

The authors compared four different voices of conversational agents to investigate the effects of gender and age on the perceived quality of psychological counseling treatments. Therefore, they developed and implemented voice agents based on the procedure of a five-staged counseling model. 88 participants, equally female and male, evaluated the session effects and characteristics of the counselors by using standardized questionnaires.  The results show that female voices ranked higher in attractiveness and older voices in expertness and depth regardless of gender. As the use of chatbots and conversational agents to support and improve personal health increases, this study helps to inform future design decisions in the choice of voice.

This application field is compelling and timely, as it is relevant both to users regarding access to mental health services and designers who engage in voice-based service design. The authors allocate their work in the shift of “computer-mediated communication to human-machine communication” and in the field of HRI. Unfortunately, the work lacks thorough research on the current body of research of conversational agents. Otherwise, the authors would be more careful to use the appropriate terms of voice-based interaction systems and not switch between the terms AI, Voice assistant, AI Voice, Conversational Agent, or AI CA throughout the text. That adds to the confusion and does not show an understanding of how the user interface and technology are related. Besides, AI seems to be used as a buzz-word, because I am not sure which contribution the authors seek to make to the field of AI/NLP/Machine Learning. The paragraph from line 135-138 is very superficial and not cited. Besides, the title seems to be misleading as well. I would recommend studying the work published at conferences like “Conversational User Interfaces”, “Designing Interactive Systems” or “Human-Agent Interaction” more closely, which contribute to the design implications of conversational agents beyond robots and AI systems. 

In contrast, a structured overview about what we know of counseling by chatbots like Replika, and how it is different from voice assistants, might strengthen the contribution. More studies of related work about conversational design like “One Voice Fits All? Social Implications and Research Challenges of Designing Voices for Smart Devices” by Cambre and Kulkarni or “The sound of trustworthiness: Acoustic-based modulation of perceived voice personality “ by Belin et al. would help to understand the current challenges of research and design of voice-based devices. For example, Sutton also claims to be careful to take voice as the primary factor of gender in “Gender Ambiguous, not Genderless: Designing Gender in Voice User Interfaces (VUIs) with Sensitivity”. Furthermore, social cues and their impact on social presence should be presented in a structured approach. It is necessary to look up the work of Feine, if you are not previously familiar with it, to understand what is meant by voice cues as part of social cues. In this context, the paper requires more motivation towards its focus on gender and voice as main personality traits, while other studies show the opposite. Furthermore, more insights about the beneficial characteristics of good counselors might highlight the importance of personality. Factors like “willingness to open up” support the use of voice agents in general, but, unfortunately, miss highlighting the impact of voice and gender of the human counselors. 

The psychological treatment procedure is properly explained, yet the research object needs further details as conversation flow, dialog trees and/or examples, fallbacks, and how the authors classified the voices as typically young and old.  Furthermore, the research design fails to elaborate on how age and gender as determinant design variables are associated with the counselor and the overall voice-based counseling design. I wonder how far voice could be reduced to the variables of gender and age alone, which seems to be very simplistic and does not exclude personal sound preferences that indicate empathy or active listening. For example, in their work Feine et al. show different factors like qualities, vocalization, content, and style influence the perception of voice.  It would be interesting to know which items have been used to assess the quality of the conversation. Further, the authors do not state any hypothesis they planned to evaluate but discussed the expected appreciation of older voices at the end of the article.

The overall experiment is explained precisely and I appreciate the authors following the ethical guidelines to exclude participants who need immediate help. Also, the questionnaires seem to be appropriate to assess the quality of the counseling and the impression of the counselor. Yet, I miss the research hypothesis of the authors. I would also recommend explaining the measurement items in more detail, as they can help to illustrate future design decisions. For example, the adjectives used for attractiveness in line 314 were not named before as items. Although the conclusions are supported by the results of the study, they seem to remain on the surface as design implications. The authors miss the opportunity to clearly state a sound contribution concerning the current state of the art.  

This study has some merit to illustrate the effects of voice styles in mental health counseling as a promising application field. At this point, it seems to be an early stage of work that needs to better position itself in the current state of the art. Hence, the contribution remains unclear and a little weak regarding prior studies on this matter. Moreover, I would recommend to proof-read the work by an English native speaker as some parts were hard to follow. Besides, in chapter 2.1 the citation style differs from the rest of the text. I think this kind of work has some potential in the field of voice-agent based counseling but lacks in this form a proper contribution to support designers.

Reviewer 2 Report

Weaknesses

  1. Tables 1 and 2 show the improvement provided by the proposed methods with respect to the means and standard deviations. However, this improvement would be more understandable if the results had also the original clean voice results as another reference value for an ideal case.
  2. No audio samples are provided. It would be nice to see a demonstration in order to see their method performance.
  3. To this respect, the authors should clearly compare the results obtained in this paper with the methods already published by some of them based on complexity measures, which provide much better accuracy. This is important to establish a fair comparison with the previous works, which up to date are considered a potential baseline. Otherwise, the conclusions extracted in this paper are quite limited and would require a new publication.
  4. All the references support the ideas proposed in the text and they are used where they are needed.
  5. The conclusions extracted in this paper are quite limited.
  6. There is not enough detail in order to replicate the study.

Reviewer 3 Report

This is a nice but relatively low-level study analizing the influence of CA voice parameters on user perception in a simulated psychotherapeutic session. The results are sound and a conditionally useful beyond this particulare context and culture. I would suggest that a couple of sentences are introduced into the text mentioning more pragmatic aspects of a nature-language conversation such as the prosody of voice and the use of methaphorical expressions. Otherwise, it is difficult to evaluate the real range of problems on the way to communication between human persons and artificial agents.
